# Few-cycle laser driven reaction nanoscopy on aerosolized silica nanoparticles

Philipp Rupp[1,2], Christian Burger[1,2], Nora G. Kling[1,2], Matthias Kübel[1,2], Sambit Mitra[1,2], Philipp Rosenberger[2], Thomas Weatherby [2,3], Nariyuki Saito [4], Jiro Itatani [4], Ali S. Alnaser[5], Markus B. Raschke[6], Eckart Rühl [7], Annika Schlander[8], Markus Gallei[9], Lennart Seiffert[10], Thomas Fennel [10,11], Boris Bergues[1,2]* & Matthias F. Kling [1,2]*

Nanoparticles offer unique properties as photocatalysts with large surface areas. Under irradiation with light, the associated near-fields can induce, enhance, and control molecular adsorbate reactions on the nanoscale. So far, however, there is no simple method available to spatially resolve the near-field induced reaction yield on the surface of nanoparticles. Here we close this gap by introducing reaction nanoscopy based on three-dimensional momentum-resolved photoionization. The technique is demonstrated for the spatially selective proton generation in few-cycle laser-induced dissociative ionization of ethanol and water on $SiO_2$ nanoparticles, resolving a pronounced variation across the particle surface. The results are modeled and reproduced qualitatively by electrostatic and quasi-classical mean-field Mie Monte-Carlo ($M^3C$) calculations. Reaction nanoscopy is suited for a wide range of isolated nanosystems and can provide spatially resolved ultrafast reaction dynamics on nanoparticles, clusters, and droplets.

[1] Max Planck Institute of Quantum Optics, D-85748 Garching, Germany. [2] Physics Department, Ludwig-Maximilians-Universität Munich, D-85748 Garching, Germany. [3] Physics Department, Technical University Munich, D-85748 Garching, Germany. [4] The Institute for Solid State Physics, The University of Tokyo, Kashiwa, Chiba 277-8581, Japan. [5] Department of Physics, American University of Sharjah, Sharjah POB26666, UAE. [6] Department of Physics, Department of Chemistry, JILA, and Center for Experiments on Quantum Materials, University of Colorado, Boulder, Colorado 80309, USA. [7] Physical Chemistry, Institute for Chemistry and Biochemistry, Freie Universität Berlin, D-14195 Berlin, Germany. [8] Macromolecular Chemistry Department, Technical University Darmstadt, D-64287 Darmstadt, Germany. [9] Chair in Polymer Chemistry, Saarland University, D-66123 Saarbrücken, Germany. [10] Institute for Physics, Rostock University, D-18051 Rostock, Germany. [11] Max Born Institute, D-12489 Berlin, Germany. *email: boris.bergues@mpq.mpg.de; matthias.kling@lmu.de

Nanomaterials exhibit a characteristic optical response, dependent on their size, material, composition, and environment[1-3]. They feature a large surface to volume ratio and catalyze chemical reactions[4], including for instance, in atmospheric photochemistry[5,6]. The concentration and enhancement of electromagnetic fields on the nanoscale is important for many applications including detection of trace substances[7], single-molecule spectroscopy and microscopy[8,9], as well as nanofocusing and modification of surfaces beyond the optical diffraction limit[10,11]. Isolated nanograins, nanoice, and other nanoparticles are known to play an important role in astrochemistry[12], enabling the (irradiation-induced) formation of complex molecules and molecular ions[13]. How these formation processes are influenced by the morphology of the nanosurfaces is, however, largely unknown and strongly motivates experimental progress in this area. In all of these applications, the nanoscale, light-induced near-fields play a critical role.

Electron emission and scattering in strong laser fields has been shown to provide nanometer-resolved information about light-induced near-fields, by mapping of the local near-fields onto the final electron momentum distributions[14-18]. Electron emission in extreme ultraviolet fields even permits sampling the near-field with sub-cycle (attosecond) temporal resolution[19,20]. Despite this progress, unraveling the impact of near-fields on photo-induced reaction yields for molecular adsorbates remains challenging[21]. In this work, we provide a solution by implementing reaction nanoscopy, which permits accessing the nanoscale reaction yield landscape via a three-dimensional momentum spectroscopy of charged molecular fragments, which beyond applications in strong-field laser physics may open up opportunities in the fields of atmospheric and astrochemistry.

In our proof-of-principle studies, we investigate proton emission from dissociative ionization of ethanol and water molecules adsorbed on $SiO_2$ nanoparticles. We find that the anisotropic proton momentum distribution measured in our experiment maps out the spatial variability of the reaction yield on the particle surface, which itself correlates with the near-field amplitude on the surface of the particle. What is denoted by near-field in the following is the sum of incoming and Mie scattered laser fields in the vicinity of the nanoparticles surface. The experimental results are modeled by semi-classical Monte-Carlo trajectory simulations[14], including Mie near-fields, molecular ionization, and charged particle interactions. Laser-generated ions from isolated nanoparticles have been studied before, to probe plasma generation in high-intensity laser fields and to provide nanoscale information about the creation of the plasma[22]. In the present work, much lower intensities are employed, yet with pulse durations of only a few optical cycles, which suppresses plasma formation[23] and the expansion of the particle during the interaction with the laser field. During that interaction, molecules on the nanoparticle surface may undergo dissociative ionization. The charged molecular fragments are emitted from the surface and serve as a sensitive probe of the local light-induced reaction yield.

## Results

**Experimental results**. Laser-generated charges are detected in a reaction nanoscope (Fig. 1), an adaptation of reaction microscopy[24] to nanotargets. Details of the setup are described in the Methods section. Briefly, linearly polarized laser pulses with a central wavelength of 720 nm, an energy of 300 μJ, and a full width at half maximum of the temporal intensity envelope of 4 fs are generated at a repetition rate of 10 kHz in an amplified Ti: sapphire laser system (Femtopower Compact Pro HR, Spectra Physics) with subsequent spectral broadening in a hollow core fiber. A fraction of the beam is focused ($f = 12.5$ cm) to an

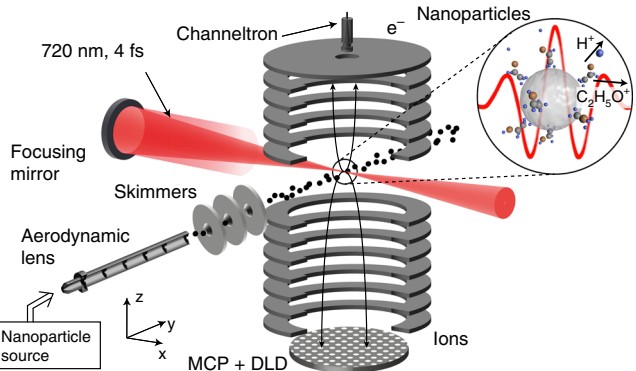

**Fig. 1** Reaction nanoscope. The nanoparticles are delivered by an aerosol generator and pass an aerodynamic lens and a set of skimmers for differential pumping. The few-cycle laser pulses cross the focused nanoparticle beam in the center of the reaction nanoscope. The $SiO_2$ nanoparticles and molecular surface adsorbates are ionized during the interaction. Fragments arising from molecular photodissociation are accelerated towards the ion detector (bottom: microchannel plates (MCP) and delay-line detector (DLD)) by a homogeneous electric field. Electrons are accelerated towards the opposite side of the spectrometer and are detected with a channeltron (top). Electrons and ions are recorded in coincidence

intensity of $\sim 5 \times 10^{13}$ W cm$^{-2}$ in the interaction region of the reaction nanoscope (cf. Fig. 1), which permits recording both ions and electrons, resulting from the interaction of light pulses with a jet of free particles, in coincidence.

The ionization of background gas produces by itself a low electron rate at the channeltron, see red curve in Fig. 2a for a measurement of a target consisting of solvent without nanoparticles. The background gas in this case consists of argon with traces of residual solvent ethanol/water molecules. In contrast, the nanoparticle ionization gives rise to a much higher and well-discriminated electron signal, as seen from the blue curve in Fig. 2a. A high electron count measured in coincidence with the ion momenta is therefore a distinct marker to identify nanoparticle ionization events, which occur in only 0.3% of all laser shots. The main contribution to the ion time-of-flight (TOF) spectrum obtained for nanoparticle hits (Fig. 2b) results from solvent molecules adsorbed on the nanoparticle surface, in this case mostly $C_2H_5OH$ (46 u), which mainly fragments into $H^+$, $CH_3^+$, $CH_2OH^+$, and $C_2H_5O^+$, and some traces of $H_2O$ (18 u), which fragments into $H^+$ and $OH^+$. The peak intensity in the focus ($8 \times 10^{13}$ W cm$^{-2}$) is determined from the $Ar^{2+}/Ar^+$ yield ratio with an estimated accuracy of 20%[25].

Careful inspection of the average TOF spectrum recorded for nanoparticle hits reveals a sensitive dependence of the $H^+$ peak to the presence of nanoparticles, apparent by the appearance of two satellite peaks in the momentum along the polarization direction ($p_{pol}$, see blue curve in inset of Fig. 2b). We note that peaks for higher masses in the TOF spectrum do not permit to resolve this feature due to the low momentum difference. The TOF spectra indicate that the protons are mainly generated from the dissociation of water or ethanol molecules (or to some extent also silanols) on the nanoparticle surface.

We have carried out experiments for $SiO_2$ nanoparticles with a diameter of $d = 110$ nm and $d = 300$ nm. Selecting the events that are coincident with a high electron signal facilitates the efficient suppression of the proton signal from the background gas. For both particle sizes, the final proton momentum distribution cannot be explained by strong-field dissociative ionization of ethanol or water alone[26]. Indeed, protons from the background

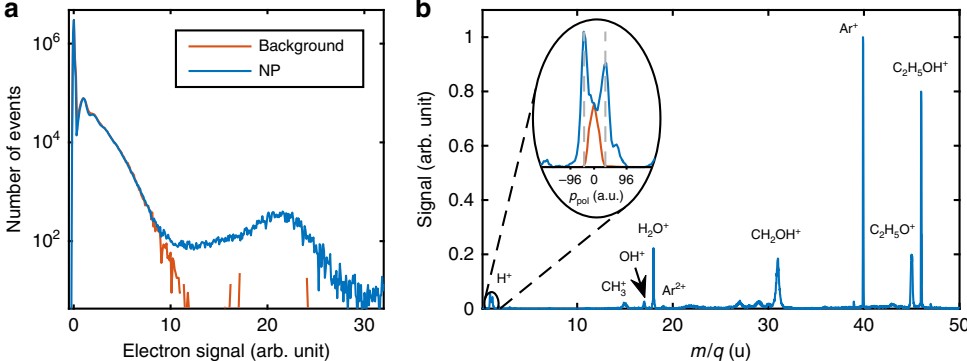

**Fig. 2** Experimental data. **a** Histogram of the number of detected electrons from the interaction of few-cycle pulses with background gas only (red) and with 110 nm $SiO_2$ particles (blue). **b** Average ion time-of-flight spectrum of shots containing nanoparticle hits on a mass/charge ($m/q$) axis. The indicated ionic fragments arise from ionization of argon and dissociative ionization of ethanol and water. The inset shows the enlarged peak of $H^+$ on a momentum scale along the polarization direction ($p_{pol}$), for events with $SiO_2$ particles (blue) and with just background gas (red). The gray dashed lines indicate a momentum of ± 40 a.u. The $Ar^{2+}$ peak is just indicated but is not visible on a linear scale

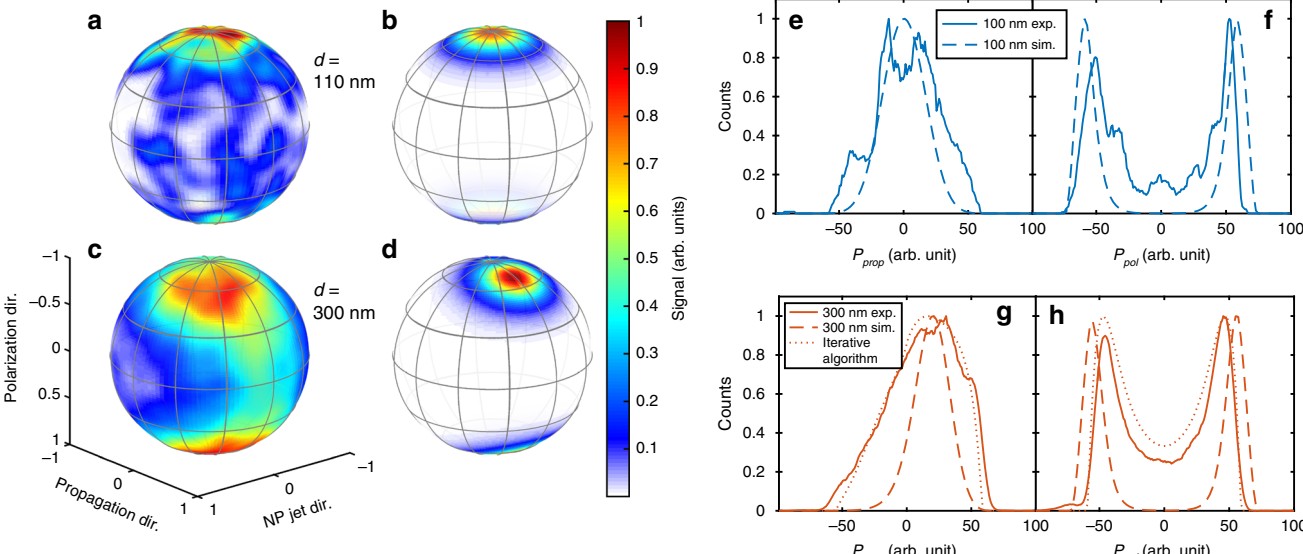

**Fig. 3** Comparison of measured and simulated proton distributions. In **a**–**d**, the 3D ($\varphi, \theta, r$) momentum distributions of protons are integrated along the radial coordinate and the retrieved two-dimensional ($\varphi, \theta$) density map is spanned over a unit sphere. The coordinates $\theta$ and $\varphi$ are defined in Fig. 5 and a detailed description of the projection is given in the Supplementary Note 1. The number of protons per solid angle is encoded in the color scale. **a** Measured and **b** simulated distribution for the 110 nm particles. **c** Measured and **d** simulated distribution for the 300 nm particles. **e** Comparison of the measured momentum distribution along the propagation direction (solid blue line) with the simulated distribution (dashed line) for the 110 nm particles. **f** Same comparison along the polarization direction. **g**, **h** Same as **e**, **f** but for the 300 nm particles. The dotted lines correspond to the retrieved dissociation yield distributions

gas, which are generated in the absence of nanoparticles, have a narrower momentum distribution with a single peak at zero momentum (cf. red curve in inset of Fig. 2b). We infer from this comparison that energetic protons in the nanoparticle experiments originate from molecular dissociative ionization on the nanoparticle surface. The strong dependence of the observed proton momentum distribution on the nanoparticle size corroborates this hypothesis. As seen in Fig. 3, the angular proton distribution has a dipolar shape for 110 nm particles (Fig. 3a), whereas it exhibits a strong asymmetry for 300 nm particles (Fig. 3c). This distribution correlates with the expected intensity distribution of the laser-induced near-fields for the investigated nanoparticles. For particle sizes that are small compared with the wavelength, the spatial distribution of the near-field intensity has a dipolar shape, whereas for particle sizes approaching the wavelength, the maximum of the distribution bends forward in the light propagation direction[14].

The measured momentum distributions are shown as radial projections (Fig. 3a, c) and as projections onto the propagation axis (solid line in Fig. 3e, g) and polarization axis (solid line in Fig. 3f, h).

**Theoretical results**. To explore the mechanisms responsible for the experimental proton momentum distributions, we performed three-dimensional semi-classical $M^3C$ (mean-field Mie Monte-Carlo) simulations of the charged particle (electrons and ions) dynamics (see Methods for details). In our model, electrons are liberated via tunnel ionization, under the action of the total electric field consisting of the laser field and the induced field of the sphere resulting from bound and free charges. The field is described using a self-consistent two-level scheme, where the linear contribution is treated via the Mie solution and the correction that includes all nonlinear contributions is described in

quasistatic mean-field approximation. At each time step, a Monte-Carlo method is used to launch electron trajectories weighted according to an Ammosov–Delone–Krainov-type tunneling rate[27]. Elastic and inelastic collisions of electrons with the nanoparticle are included in the propagation[14].

In addition to ionization and propagation of electrons, we simulate the yield for the dissociative ionization (see Methods) and calculate the trajectories of protons emerging from the strong-field dissociation of solvent molecules adsorbed on the nanoparticle surface. The calculation results for 110 and 300 nm particles (Fig. 3b, d and dashed lines in Fig. 3e–h) reproduce the characteristic trend of the experimental observations, i.e., the presence of pronounced directional emission hot spots and their movement in propagation direction towards the back side of the particle with increasing diameter. This trend is expressed most clearly in the peak shift of the projected proton momentum distributions. In the following sections, we show how the anisotropic dissociation yield induced on the nanoparticle surface by the near-field is mapped onto the final proton momentum distribution. Based on the good qualitative agreement, we use the simulations to disentangle the different effects leading to the observed momentum distributions.

## Discussion

Earlier work on the interaction of few-cycle pulses with nanoparticles has concentrated on the mechanism of electron acceleration after photoemission from a solid[14]. Süßmann et al.[14] have revealed that electrons are generated on the nanoparticle surface in the regions of maximum field enhancement and subsequently accelerated in the local near-fields. It has been shown experimentally and theoretically that released electrons gain most of their final energy from a combination of the dielectrically enhanced laser field and a local trapping potential induced by ionization[14,15,28].

In contrast, in the present study on molecular adsorbates, we find that the much heavier protons do not gain significant energy by the enhanced field around the nanoparticle (see Fig. 4, inset). The M³C simulations indicate that the final proton momenta are mainly determined by the electrostatic field of the charged nanoparticle (cf. Fig. 4).

The electrostatic field arises from released electrons and the bound ions in the nanoparticle[29]. The effective field is repulsive for the protons. Coulomb attraction between the fast escaping electrons and the ions created on the nanoparticle is negligible and has no significant effect on the final proton momenta. In contrast, the proton dynamics are dominated by electrostatic interactions with the positively charged nanoparticle surface. These charges form an inhomogeneous surface potential that traps electrons in a layer close to the surface and screens the inside of the nanoparticle[29]. Owing to the highly nonlinear nature of the light–matter interaction, both the ionization of the nanoparticle and the dissociation of the molecules on its surface are temporally confined to the laser pulse duration and occur preferentially in the regions with the largest total field strengths. Their distribution resembles the shape of the H⁺ momentum distribution: a dipolar shape for 110 nm particles and the asymmetric structure for 300 nm particles (cf. Fig. 3). In contrast to the much faster emission of electrons, protons efficiently probe the nanosphere surface on a picosecond time scale, which facilitates a mapping between the dissociation yield landscape on the surface and the final momentum distribution. We note here that this is a major difference compared with earlier work[14] and forms the basis for the mapping of reaction yields in the reaction nanoscope.

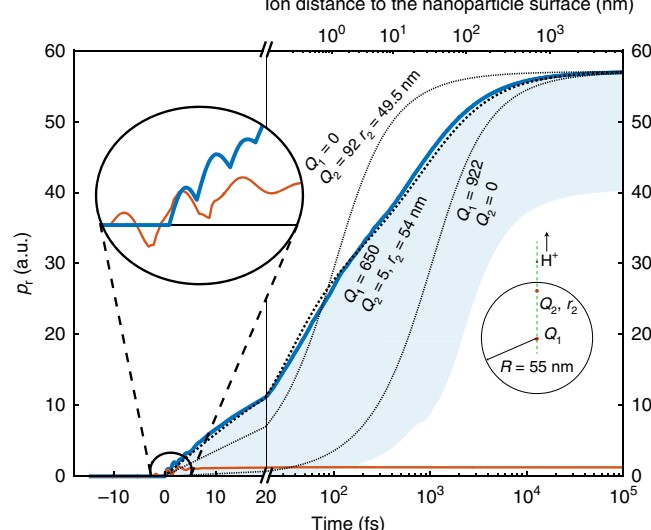

**Fig. 4** Analysis of proton and electron trajectories. The M³C simulations are performed for 110 nm SiO₂ particles. The radial momentum $p_r = |\vec{p}|$ of the cutoff electrons (red line) represents an average over the 10% highest electron momenta. The radial proton momentum (blue line) is calculated for a proton released from the surface at the pole ($\theta = \pi/2$) of the particle. The blue shaded region represents the spread in the velocity gained by protons released at different positions on the nanoparticle. The axis at the top indicates the distance of the proton from the nanoparticle surface at the respective times shown at the bottom. The left inset is a magnification of the region where the dynamics is laser-field driven. The right inset is an illustration of a simple model describing the 1D trajectory of a proton in the static field of two point charges, representing the (asymmetric) surface potential. The three dotted lines in the main graph show the trajectories for the model parameters indicated in the inset

The essence of the proton dynamics can be captured with a one-dimensional (1D) model along a radial axis (see green dashed line in the sketch in Fig. 4), where the electrostatic repulsion from two positive point charges is considered. A description by two point charges reflects the initial asymmetric surface potential around the sphere. A first charge $Q_1$ is situated in the center of the sphere and another charge $Q_2$ is placed at $r_2$, below the nanosphere surface. A proton is launched from the surface on the axis defined by $Q_1$ and $Q_2$. Three free parameters (charges $Q_1$, $Q_2$, and radius $r_2$) in total are enough to reproduce the correct radial dynamics of a proton in the field of the anisotropically charged particle (see Methods for details). A large charge $Q_1$ in the center is necessary to model the correct final momentum, whereas the second charge $Q_2$ at radius $r_2$, introduced to represent the asymmetry in the charge distribution, ensures good agreement in the dynamic behavior, cf. Fig. 4.

The different time scales of the process allow the separation of the dynamics into two phases: a first phase, occurring on femtosecond scales, during which the surface charge distribution and the probe charges are generated in the laser-induced near-field, and a second phase, occurring on the picosecond scale, in which the probe charges are accelerated away from the now charged nanoparticle surface.

The above analysis suggests that the spatially resolved reaction yield on the surface can be retrieved from the measured proton momentum distributions. In order to solve this inverse problem, we use our simulations to provide a more quantitative description of the mapping between the initial proton position and the final proton momentum.

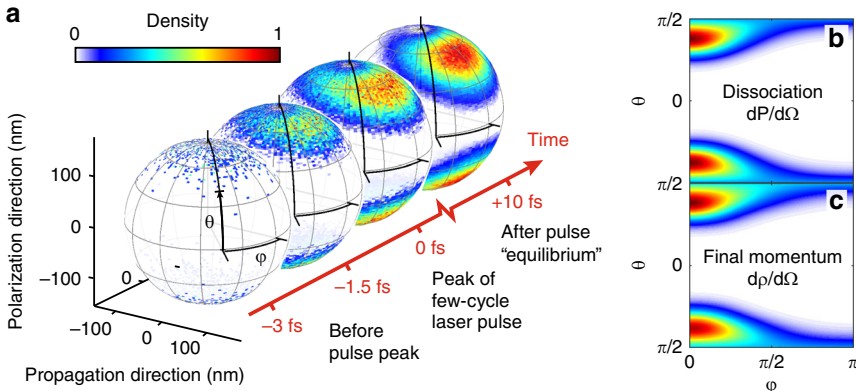

**Fig. 5 Dissociation yields on the nanoparticle surface. a** Time evolution of the surface charge distribution simulated for the 300 nm particle at a single intensity and averaged over the carrier-envelope phase (CEP). Two snapshots are shown during the rising edge of the laser pulse, one at the peak electric field and one 10 fs after the interaction with the laser pulse. Each point on the sphere is defined by the elevation angle $\theta$ and the azimuthal angle $\varphi$, in the intervals $\left[-\frac{\pi}{2}; \frac{\pi}{2}\right]$ and $[0; 2\pi]$, respectively. The angle $\theta$ is measured with respect to the propagation/NP-jet-plane and $\varphi$ is measured with respect to the propagation axis. The angle $\varphi$ is only shown from 0 to $\pi$ from now on due to the mirror symmetry with respect to the polarization–propagation plane. **b** Differential probability distribution $dP/d\Omega$ for the deprotonation reaction as a function of $\theta$ and $\varphi$. **c** Experimentally accessible momentum distribution of the final proton momenta as a function of $\theta$ and $\varphi$. All distributions or rates are normalized to a maximum value of 1 and use the shown color scale

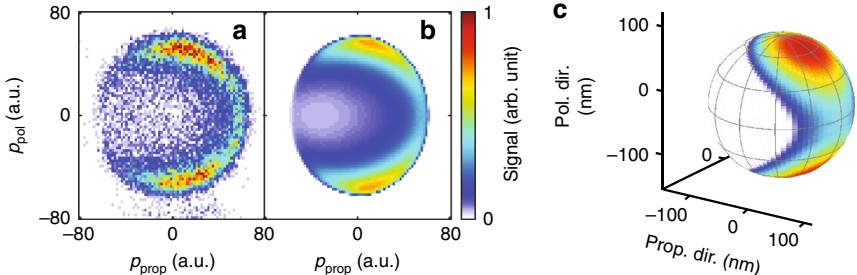

**Fig. 6 Retrieval of the dissociation yield distribution. a** Proton momentum distribution measured in the experiment with the 300 nm particle, projected onto the polarization and propagation plane. **b** Same projection of the retrieved momentum distribution (see text). **c** The retrieved surface charge density visualized on the surface of a nanoparticle

The dynamics of the charge distribution on the surface of the nanoparticle calculated using the M³C code is shown in Fig. 5a. The spherical particle is ionized by the laser pulse on a time scale of a few femtoseconds. After this initial laser interaction, the surface charge distribution reaches a quasi-equilibrium, which is governed by the local near-fields around the nanoparticle. Similarly, the initial spatial distribution of protons, which is shown in Fig. 5b as a function of the elevation and azimuthal angles $\theta$ and $\varphi$, depends on the electric field strength. The resulting final proton momentum distribution is depicted in Fig. 5c. For the parameters of our experiment, the two distributions are almost indistinguishable, stressing the close relation between position–space and momentum–space. We show in the Supplementary Note 2 (Supplementary Figs. 2 and 3), that for the parameters of the present experiment, a later relaxation of the surface charges into an isotropic distribution does not affect the mapping between yield distribution and final momentum distribution. The question of the existence and time scales of transient charge relaxation processes will have to be elucidated in future studies.

In order to retrieve the spatially resolved dissociation yield, we have implemented an iterative optimization procedure that minimizes the deviation between the measured and the calculated momentum distribution. In the algorithm, which is described in the Methods section, the reaction yield and the surface charge distribution are represented by a linear combination of spherical harmonics and the expansion coefficients are varied together with the nonlinear order of the dissociative ionization process. The retrieved set of optimized parameters is in qualitative agreement with the charge distribution and dissociation rate calculated using full M³C simulations (see dotted lines in Figs. 3g, h and 6). The use of this iterative optimization procedure is not limited to the parameters of the present experiment. As shown in the Supplementary Information, it provides a general framework for inverting the yield-to-momentum mapping and works for more complex cases, where the relation between the initial yield distribution and the final momentum distribution is much more intricate.

Our results show that protons from the dissociation of molecular adsorbates on nanoparticle surfaces can serve as a probe for both the surface charge distribution induced by the near-field of nanospheres and the resulting spatially dependent dissociative ionization yield. Qualitative agreement with the experimental data is obtained from semi-classical simulations that incorporate the near-field, the rate of the dissociative ionization, and many-particle charge interactions. We find that the mapping of reaction yield to the final proton momentum takes place after the interaction and is governed by the electrostatic field of the charged nanoparticle alone. This enables the reconstruction of the nanoscale reaction yield landscape from the measured data. The reaction nanoscope can open the door for the spatially resolved study of nanoparticle photochemistry including

its spatio-temporal variation in time-resolved pump–probe implementations.

## Methods

**Experimental setup.** The nanoparticle source has been described in detail in refs. [30,31]. Briefly, the nanoparticles, dispersed in ethanol or water, are aerosolized using a fast argon gas stream. Nanoparticle clusters are eliminated from the gas stream by an impactor unit. A reverse-flow dryer is used to control the amount of solvent molecules on the nanoparticle surface. An aerodynamic lens focuses the nanoparticle beam to a spot size of ~0.5 mm in diameter in the center of the reaction nanoscope, where ultra-high vacuum ($10^{-9}$ mbar) conditions are maintained. Electrons and ions created in the interaction region are accelerated within a homogeneous electrostatic field (150 V/cm) towards their respective detectors (see Fig. 1). The electron side is equipped with a channeltron, enabling counting the number of released electrons. A calibration of the channeltron was performed by increasing the background pressure in the interaction chamber and monitoring the increase in electron signal. Ions are detected with a time- and position-sensitive detector consisting of a multichannel plate and a delay-line detector. From the TOF and position, the three-dimensional initial momenta of the fragment ions are retrieved. The data for electrons and ions are collected in coincidence for each laser shot up to the full repetition rate of 10 kHz. To preserve coincidence conditions, the total count rate in the experiment is maintained at ~0.3 ionization events per shot, resulting in about 30 laser–nanoparticle interactions per second due to the dilute nanoparticle beam.

**Nanoparticle preparation.** Silica nanoparticles with diameters of 110 and 300 nm, and a narrow size distribution were prepared by wet chemistry approaches. First, small seed nanoparticles were prepared by the Stöber method[32]. In a typical seed preparation procedure, 21 g of TEOS, 28 mL of ammonia solution (25 wt% in water), and 1 mL of water were added to 530 mL of ethanol and stirred for 12 h. A further shell was grown on the silica nanoparticles by the seeded growth method[33] until the desired particle size was reached. All samples have been stored in ultra-pure ethanol after cleaning. Characterization by transmission electron microscopy and dynamic light scattering yielded a polydispersity of about 4.9% for the 110 nm and 2.9% for the 300 nm particles, respectively. The surface of silica nanoparticles prepared by the Stöber method are typically covered by silanols, i.e., Si-OH groups[34].

**Simulation details.** The electron trajectories are calculated using the M³C-code described in refs. [14,15], which has shown to quantitatively agree with measured electron momentum distributions for few-cycle ionization of nanoparticles[14]. Protons are released at the peak of the laser pulse with zero initial momentum. We assume an $I^n$ intensity dependence for the dissociation rate to account for the spatial dependence of the ionization probability at the nanosphere surface. Here we choose $n$ such that the molecules can be ionized. The ionization potentials are 12.6 eV for water[35] and 10.5 eV for ethanol[36]. The additional energy required for dehydrogenation of the molecules at local near-field intensities reaching $1.2 \times 10^{14}$ W cm$^{-2}$ (with a near-field intensity enhancement for 300 nm spheres of up to 2.8) is assumed to be contributed from laser-driven (re)scattering of electrons. The classical maximal recollision energy would yield 18.0 eV ($=3.2\,U_p$). This is sufficient for H$^+$ formation in the cation of the species. Previous studies report H$^+$ formation from water cations with 6.2 eV[37]. For ethanol, the H$^+$ formation from the cation may be inferred from data on other hydrocarbons, such as hydroxymethyl groups with 7.3 eV above the cation ground state[38]. This yields $n \approx 10.9$ and $n \approx 10.3$ for water and ethanol, respectively. With the nanoparticle near-fields, these channels can be reached. Without near-field enhancement, the maximal recollision energy is only about 9 eV and H$^+$ formation is strongly suppressed, which is consistent with our data.

An adaptive time-step scheme is used to facilitate the propagation of protons up to 3 ns, where we find the momenta to be converged. Averaging over the intensity distribution in the focal volume is taken into account assuming a Gaussian beam profile. Low intensities leading to a low number of electrons are neglected to resemble the experimental analysis. The number of detected electrons in the simulations account for the geometric constraints given by the size of the channeltron and the detection efficiency of the channeltron. The laser intensities that lead to a small electron signal are comparably low and only affect the very central part of the momentum distribution.

**Analytical 1D model.** If only one of the charges $Q_1$ or $Q_2$ is taken into account and the other charge is set to zero, the model represents the charge distribution on small nanoparticles and the equations of motion for this simplified situation can be derived analytically. It assumes a positive charge $Q$ at position $r = 0$ for all times. The positive probe charge $q$ of mass $m$ is at position $r(t = 0) = R$ with $v(t = 0) = 0$. The equations of motion are solved in one dimension and result in:

$$t = \sqrt{\frac{2\pi m \varepsilon_0}{Qq}}\left[\sqrt{Rr(r-R)} + R^{3/2}\log\left(\sqrt{\frac{r}{R}-1} + \sqrt{\frac{r}{R}}\right)\right], \quad (1)$$

$$p = \sqrt{\frac{mQq}{2\pi\varepsilon_0}\left(\frac{1}{R} - \frac{1}{r}\right)}. \quad (2)$$

The two characteristic quantities are

$$p_f(t \to \infty) = \sqrt{\frac{mQq}{2\pi\varepsilon_0 R}} \sim \sqrt{\frac{Q}{R}} \quad \text{and} \quad (3)$$

$$t_c = \sqrt{\frac{2\pi m \varepsilon_0 R^3}{Qq}}\left(\sqrt{2} + \log\left(1+\sqrt{2}\right)\right) \sim \sqrt{\frac{R^3}{Q}} \text{ with } t_c \text{ defined by } E(t_c) = 0.5E(t \to \infty) \quad (4)$$

The only free parameters here are the position $R$ and the number $Q$ of elementary charges. The final momentum is determined by the ratio $p_f \sim \frac{Q}{R}$. The number of charges ($Q \approx 920$) obtained from the analytical model for the measured final momentum of ~55 a.u. is comparable to that obtained with the numerical M³C simulations. However, an accurate fit of the temporal dynamics predicted by the M³C simulations with the simple model requires the inclusion of two charges (see central dotted line in Fig. 4 for this scenario). Here, the trajectory is calculated numerically by integrating the differential equations. The position of the first charge is fixed to $r_1 = 0$, whereas the second position $r_2$ and the charge amounts $Q_1$ and $Q_2$ are used as fit parameters. The fit of the two-charge model to the full M³C simulations reveals a slightly reduced charge in the center ($Q_1 \approx 650$) and a very small charge ($Q_2 \approx 5$) located just 1 nm below the surface.

**Iterative optimization algorithm.** For a completely spherically symmetric charge distribution, all protons are pushed away radially from the nanoparticle. In that case, the final momentum direction coincides with the direction of the initial position vector of the proton and the final angular proton distribution is identical to the initial angular distribution of the dissociative ionization yield. In reality, the polarization direction, as well as propagation effects, break the spherical symmetry of the initial charge distribution on the nanoparticle and the mapping departs from the identity. A higher charge density in certain regions leads to a larger accelerating Coulomb force and thus a larger final momentum. At the same time, the anisotropic charge distribution accelerates the protons also tangentially to the surface, which effectively alters their direction in the $(\theta, \varphi)$-plane. As visible in Fig. 5b, c and discussed in the Supplementary Note 2, these distortions have a negligible effect in the present experiment, but do play a significant role for more complex surface charge distributions. For arbitrary charge distributions, however, the retrieval of the initial proton densities from the measured distribution is not straightforward.

The mathematical description of this retrieval problem is as follows: the surface charge distribution is approximated by spherical harmonics $Y_l^m(\theta, \varphi)$ with order $l = [0;L]$. Due to the fact that the distribution is real valued and the plane of symmetry (polarization–propagation plane), we need $L(L+1)$ coefficients. For simplicity, we assume that the H$^+$ density $\rho_{H^+}$ on the surface scales as $|E(\theta, \varphi)|^n$, where $E$ is the electric field created by the surface charges and $n$ is used as an additional fitting parameter. The electrostatic field and the H$^+$ density are then calculated from the nanoparticle surface charges and a trajectory analysis yields the final proton momenta. The Nelder–Mead simplex fitting algorithm is used to optimize all $L(L+1)+1$ variables to minimize the deviation between the measured and reconstructed momentum distributions shown in Fig. 6a, b, respectively. This iterative optimization algorithm enables the reconstruction of the dissociation yield and charge distribution on the nanoparticle surface from the measured proton distribution in Fig. 3 (see Fig. 6).

## Data availability

The data that support the findings of this study are available from the corresponding authors upon reasonable request.

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

## Acknowledgements

We are grateful for support by the DFG through SPP1840 (L.S., T.F., and M.F.K.), SFB 652/3 (L.S. and T.F.), LMUexcellent (P. Rupp, C.B., N.G.K., and M.F.K.), and the Cluster of Excellence: Munich Centre for Advanced Photonics (P. Rupp, C.B., N.G.K., B.B., M.F. K.). P. Rupp, C.B., M.K., and M.F.K. acknowledge support from the EU via the ERC grant ATTOCO (number 307203). A.S. and M.G. acknowledge partial support in the frame of the DFG project GA-2169/5-1. C.B. and S.M. acknowledge support from the Max Planck Society via the IMPRS-APS.

## Author contributions

P. Rupp, C.B., and N.G.K. contributed equally to this work. M.F.K. conceived the idea for the experiment. P. Rupp, C.B., M.K., S.M., T.W., P. Rosenberger, and N.G.K. developed the experimental setup and performed the measurements. N.S. and J.I. contributed to the intensity calibration procedure. A.S. and M.G. prepared and characterized the $SiO_2$ nanoparticles, and discussed the nanoparticle synthesis. E.R. contributed to the design of the aerosol source. P. Rupp, T.W., and B.B. developed the simulation model and performed the electrostatic (T.W. and B.B.) and Monte-Carlo simulations (P. Rupp), which are based on the $M^3C$ code developed by L.S. and T.F. P. Rupp, C.B., N.G.K., M.K., S.M., P. Rosenberger, T.W., B.B., and M.F.K. evaluated, analyzed, and, together with A.S.A. and M.B.R., interpreted the results. All authors discussed the results and contributed to the final manuscript drafted by P. Rupp, B.B., and M.F.K.

## Competing interests

The authors declare no competing interests.
