## [Peer Review File · Nature Communications]

Reviewers' comments:

Reviewer #1 (Remarks to the Author):

In this manuscript "Few-cycle laser driven reaction nanoscopy on isolated nanoparticles", Philipp Rupp et al. performed a novel experiment to map out the spatially selective proton generation on SiO₂ nanoparticles with few-cycle laser induced dissociative ionization of ethanol and water. Their main supportive evidences for this conclusion come from (1) Experimental demonstration of detected electrons and ionic fragments from the interaction of the pulses (2) Comparison of measured and simulated results, which yields the ultrafast surface charge distribution for a 300 nm particle. In my opinion, the current work is an important experimental observation and the authors have not doubly demonstrate their ability to perform both high spatial and time resolution, which is impressive. In principle I can recommend publication in Nature communication after a few points been clarified, listed below:

1. The conclusion is not clear. The authors should discuss a little bit more about the applications of their techniques either in introduction or summary.
2. In line 51-52, "following the near-field distribution at the particle", at this stage the readers are not sure what does the author mean by "near-field distribution"
3. "Molecular fragments" need a better definition for the non-experts.
4. It is not easy for the reader to follow how the map in Fig. 3 (the angular proton distribution) is obtained, a schematic of the extraction process might be helpful for the discussion.
5. Can the authors also discuss what happens at a longer time scale for the dissociation on the nanoparticle surface? (Fig.5)

Reviewer #2 (Remarks to the Author):

The authors claim that they demonstrated the technique for the spatially selective proton generation in few-cycle laser-induced dissociative ionization of ethanol and water on SiO₂ nanoparticles. The results were analyzed by electro-static and M³C calculation in a quasi-classical way. The approach as reaction nanoscopy is interesting, but unfortunately, more data are required to show the validity of it. For example, they claim a good quantitative agreement between measured and simulated angular proton distributions for 110 nm and 300 nm particles. However, a clearer explanation related to Fig. 3, and more explicit comparison data should be supplied. More evidence/data should be presented in the manuscript to strengthen their conclusion (Lines 229~233). The current manuscript, therefore, should not be recommended for the publication to the journal, before answering the above comments and the following questions.

(1) Are photodissociative ionization of ethanol/water molecules on SiO₂ particles really induced by light near fields? They use 4 fs laser pulse illumination with an intensity of $\sim 5 \times 10^{13}$ W/cm², which can generally produce nonlinear processes such as multi-photon absorption and dissociation. Explicit citation of dissociation/ionization energies of the molecules will be useful to distinguish near-field effect from nonlinear ones as well as the dissociation mechanism.

(2) They model the phenomenon by using effective charges and their static potential only, but near-field transient phenomena are requested to be carefully analyzed by dynamic field treatments for quantitative discussion.

(3) What is the physical meaning and significance to do an iterative optimization? (Lines 216~217).

Reviewer #1:

In this manuscript "Few-cycle laser driven reaction nanoscopy on isolated nanoparticles", Philipp Rupp et al. performed a novel experiment to map out the spatially selective proton generation on SiO₂ nanoparticles with few-cycle laser induced dissociative ionization of ethanol and water. Their main supportive evidences for this conclusion come from (1) Experimental demonstration of detected electrons and ionic fragments from the interaction of the pulses (2) Comparison of measured and simulated results, which yields the ultrafast surface charge distribution for a 300 nm particle.

In my opinion, the current work is an important experimental observation and the authors have no doubly demonstrate their ability to perform both high spatial and time resolution, which is impressive. In principle I can recommend publication in Nature communication after a few points been clarified, listed below:

1. The conclusion is not clear. The authors should discuss a little bit more about the applications of their techniques either in introduction or summary.

We have added the two following sentences in the introduction to indicate where the technique may help elucidating open questions:

"Isolated nanograins, nanoice, and other nanoparticles are known to play an important role in astrochemistry [1], enabling the (irradiation-induced) formation of complex molecules and molecular ions [2]. How these formation processes are influenced by the morphology of the nanosurfaces is, however, largely unknown."

"... which beyond applications in strong-field laser physics may open up new opportunities in the fields of atmospheric and astrochemistry."

2. In line 51-52, "following the near-field distribution at the particle", at this stage the readers are not sure what does the author mean by "near-field distribution"

We have clarified the meaning of near-field distribution in the text by adding the sentence:

"We find that the anisotropic proton momentum distribution measured in our experiment maps out the spatial variability of the reaction yield on the particle surface, which itself correlates with the near-field amplitude on the surface of the particle. What is denoted by near-field in the following is the sum of incoming and Mie scattered laser fields in the vicinity of the nanoparticles surface."

3. "Molecular fragments" need a better definition for the non-experts.

By molecular fragments, we mean the dissociation products resulting from dissociative ionization of molecules in the strong laser field. We have changed the last sentence of the introduction to clarify this term:

"During that interaction, molecules on the nanoparticle surface may undergo dissociative ionization. The charged molecular fragments are emitted from the surface and serve as a sensitive probe of the local light-induced reaction yield."

4. It is not easy for the reader to follow how the map in Fig. 3 (the angular proton distribution) is obtained, a schematic of the extraction process might be helpful for the discussion.

We have added the following figure (Fig. S11) and text to the supplementary information to explain how exactly the angular proton distribution map is obtained.

In order to directly visualize the three-dimensional proton momentum distribution $\rho(\theta, \varphi)$ (Fig. 3), we project it radially onto a unit sphere (Fig. S11). The number dN of protons detected per solid angle $d\Omega = \cos \theta d\varphi d\theta$ on this unit sphere is encoded in the color map of the surface plot:

$$\rho(\theta, \varphi) = \frac{dN(\theta, \varphi)}{d\Omega} = \frac{dN(\theta, \varphi)}{\cos \theta d\varphi d\theta}$$

The azimuthal angle φ and the polar angle θ are introduced to allow a convenient description in spherical coordinates. In that representation, the plotted proton yield is integrated over the kinetic energy.

Figure S11. Projection method. The 3D proton momentum distribution measured in the reaction nanoscope is projected radially onto a unit sphere. The proton density is encoded in the color map (Fig. 3c).

5. Can the authors also discuss what happens at a longer time scale for the dissociation on the nanoparticle surface? (Fig.5)

Owing to the fact that the proton emission in the presented experiment results from a highly non-linear interaction, the dissociation process is temporally confined to the laser pulse duration. The measured data allows us to exclude dissociation induced by the static field of the charged particle alone, since this would result in long tails in the expressed TOF-peaks, which we do not observe experimentally. We can, however, not fully exclude dissociation of the ionized molecules that could be induced by weak post-pulses on a picosecond time scale.

We have added the following sentence in the manuscript to clarify this point:

“Owing to the highly nonlinear nature of the light-matter interaction, both the ionization of the nanoparticle and the dissociation of the molecules on its surface are temporally confined to the laser pulse duration, and occur preferentially in the regions with the largest total field strengths.”

Reviewer #2 (Remarks to the Author):

The authors claim that they demonstrated the technique for the spatially selective proton generation in few-cycle laser-induced dissociative ionization of ethanol and water on SiO₂ nanoparticles. The results were analyzed by electro-static and M³C calculation in a quasi-classical way. The approach as reaction nanoscopy is interesting, but unfortunately, more data are required to show the validity of it. For example, they claim a good quantitative agreement between measured and simulated angular proton distributions for 110 nm and 300 nm particles. However, a clearer explanation related to Fig. 3, and more explicit comparison data should be supplied.

Following the suggestion of this referee, and also the first referee, we have added a figure (Fig. S11) in the supplementary information to clarify how the angular proton distribution maps presented in Fig. 3 have been obtained:

In order to directly visualize the three-dimensional proton momentum distribution $\rho(\theta, \varphi)$ (Fig. 3), we project it radially onto a unit sphere (Fig. S11). The number dN of protons detected per solid angle $d\Omega = \cos \theta d\varphi d\theta$ on this unit sphere is encoded in the color map of the surface plot:

$$\rho(\theta, \varphi) = \frac{dN(\theta, \varphi)}{d\Omega} = \frac{dN(\theta, \varphi)}{\cos \theta d\varphi d\theta}$$

The azimuthal angle φ and the polar angle θ are introduced to allow a convenient description in spherical coordinates. In that representation, the plotted proton yield is integrated over the kinetic energy.

Figure S11. Projection method. The 3D proton momentum distribution measured in the reaction nanoscope is projected radially onto a unit sphere. The proton density is encoded in the color map (Fig. 3c).

We agree with the referee that a more explicit comparison between experiment and simulation is needed to strengthen our claims. We now provide a quantitative comparison between simulation and experiment in Fig. 3, which we, together with the text, have amended with one-dimensional plots of the density distribution projected along the different coordinate axes (Fig. 3e-h):

The measured momentum distributions are shown as a radial projections (Fig. 3 a, c) and as projections onto the propagation axis (solid line in Fig. 3e, g) and polarization axis (solid line in Fig. 3f, h).

Figure 3. Comparison of measured and simulated proton distributions. In (a-d) the 3D (φ , ϑ , r) momentum distributions of protons are integrated along the radial coordinate and the retrieved two dimensional (φ , ϑ) density map is spanned over a unit sphere (a detailed description of the projection is given in the Supplementary Information). The number of protons per solid angle is encoded in the color scale. (a) Measured and (b) simulated distribution for the 110 nm particles. (c) Measured and (d) simulated distribution for the 300 nm particles. (e) Comparison of the measured momentum distribution along the propagation direction (solid blue line) with the simulated distribution (dashed line) for the 110 nm particles. (f) Same comparison along the polarization direction. (g, h) Same as (e, f) but for the 300 nm particles. The dotted lines correspond to the retrieved dissociation yield distributions.

More evidence/data should be presented in the manuscript to strengthen their conclusion (Lines 229-233).

We realize that our initial formulation (lines 229-233 of the previous version) was somewhat unclear. We have reformulated this paragraph and believe that it has gained in clarity:

Qualitative agreement with the experimental data is obtained from semi-classical simulations that incorporate the near-field, the rate of the dissociative ionization, and many-particle charge interactions. We find that the mapping of reaction yield to the final proton momentum takes place after the interaction and is governed by the electrostatic field of the charged nanoparticle alone.

In addition, we provide new simulation results in the amended Supporting Information (Fig. S14) to demonstrate that our retrieval method is not limited to a particular momentum distribution, but also works for more complex yield distributions:

Figure S14. Retrieval algorithm for complex dissociation yield distributions on a 100 nm nanoparticle. a) Surface charge distribution, b) surface yield distribution, c) final momentum distribution, d) retrieved surface charge distribution, e) retrieved surface yield. f, g) Initial (blue) and retrieved (red) surface yield distribution on the surface of the nanoparticle projected on the propagation (f) and polarization (g) axes. The yellow line in (e, f) is the projection of the final proton momentum distribution onto the propagation and polarization axis, respectively.

The possible implementation of pump-probe experiments is meant as an outlook for future research.

(1) Are photodissociative ionization of ethanol/water molecules on SiO₂ particles really induced by light near fields? They use 4 fs laser pulse illumination with an intensity of $\sim 5 \times 10^{13} \text{ W/cm}^2$, which can generally produce nonlinear processes such as multi-photon absorption and dissociation. Explicit citation of dissociation/ionization energies of the molecules will be useful to distinguish near-field effect from nonlinear ones as well as the dissociation mechanism.

We appreciate the comment by the referee and have now explicitly included ionization and H⁺ formation for the molecules in the Methods section. Furthermore, we discuss the role of the near-field on the dissociation dynamics. We added the following paragraph in the Methods section:

The ionization potentials are 12.6 eV for water [3] and 10.5 eV for ethanol [4]. The additional energy required for dehydrogenation of the molecules at local near-field intensities reaching $1.2 \times 10^{14} \text{ W/cm}^2$ (with a near-field intensity enhancement for 300 nm spheres of up to 2.8) is assumed to be contributed from laser-driven (re)scattering of electrons. The classical maximal recollision energy would yield 18.0 eV ($=3.2 U_p$). This is sufficient for H⁺ formation in the cation of the species. Previous studies report H⁺ formation from water cations with 6.2 eV [5]. For ethanol, the H⁺ formation from the cation may be inferred from data on other hydrocarbons, such as hydroxymethyl groups with 7.3 eV above the cation ground state [6]. With the nanoparticle near-fields,

these channels can be reached. Without near-field enhancement, the maximal recollision energy is only about 9 eV and H^+ formation would accordingly be strongly suppressed, in full agreement with our data.

(2) They model the phenomenon by using effective charges and their static potential only, but near-field transient phenomena are requested to be carefully analyzed by dynamic field treatments for quantitative discussion. What are the time scales for relaxation? fs? ps? ns?

While we use a static potential for the iterative optimization algorithm, we would like to point out that our M³C calculations are dynamic and do take into account transient phenomena and time dependent multi-particle dynamics. Such a dynamic treatment including the Mie near-field, the generation of surface charges and electron-electron interactions is necessary to predict the initial yield and charge density without a priori information.

The use of a static potential in the retrieval algorithm finds its justification in the analysis of the full dynamical M³C simulations, which shows that the proton propagation only depends on the charge density generated during the interaction. We have carefully checked and also discuss in the text that the fast transient phenomena that occur on time scales of a few femtoseconds, have no other impact on the final proton momenta than the generation of the initial charge distribution, which, as we find, is essentially temporally confined to the laser pulse duration.

What is not included in the calculation is the possible relaxation of the surface charge distribution into an isotropic distribution. While the exact modelling of such dynamics is beyond the scope of the present work, it is nevertheless interesting to estimate the impact that such dynamics may have on the final momentum distribution. To this end, we have performed additional simulations to investigate the effect of charge relaxation dynamics and added two figures it in the Supplementary Information (Fig.SI2 and Fig.SI3):

Figure SI2. Effect of relaxation processes. The final proton momentum distribution is calculated for different relaxation times of the surface charges distribution. (a) “No relaxation” is equivalent to the simulation in Fig. A1. (b-d) The surface charge distribution relaxes into an isotropic distribution after 5 ps, 1 ps and 0 ps the interaction with the laser pulse.

Figure SI3. Projected momentum distributions. Projection of the momentum distribution shown in Fig. SI3 onto different coordinate axis: propagation axis (a) and polarization axis (b). The different colors refer to the time at which relaxation occurs.

The new simulation data show that for the parameters of our experiment, the mapping of the surface yield distribution onto the angular proton momentum distribution is rather insensitive to possible relaxation dynamics of the transient surface charge.

For more complex charge and yield distributions, such relaxation dynamics may change the final momentum distribution. Further experimental and theoretical studies are needed to elucidate this question. We have added the following sentence in the text:

We show in the Supplementary Information (Figs. SI2 and SI3), that for the parameters of the present experiment, a later relaxation of the surface charges into an isotropic distribution does not affect the mapping between yield distribution and final momentum distribution. The question of the existence and time scales of transient charge relaxation processes will have to be elucidated in future studies.

Regarding electron recombination, we do not expect a significant effect on the proton acceleration times of a few picoseconds.

(3) What is the physical meaning and significance to do an iterative optimization? (Lines 216–217).

We believe that the additional simulations results and discussion (see Fig. SI 4 above) should have already clarified the significance of the iterative optimization for the retrieval of the yield. We have also added the following sentence in the text:

This iterative optimization algorithm enables the reconstruction of the dissociation-yield- and charge distribution on the nanoparticle surface from the measured proton distribution in Fig. 3 (see Fig. A1).

We also reformulated the description of the iterative optimization in the Methods section to make it more transparent:

For a completely spherically symmetric charge distribution, all protons are pushed away radially from the nanoparticle. In that case, the final momentum direction coincides with the direction of the initial position vector of the proton and the final angular proton distribution is identical to the initial angular distribution of the dissociative ionization yield. In reality, the polarization direction, as well as propagation effects, break the spherical symmetry of the initial charge distribution on the nanoparticle, and the mapping departs from the identity. A higher charge density in certain regions leads to a larger accelerating Coulomb force, and thus a larger final momentum. At the same time, the anisotropic charge distribution accelerates the protons also tangentially to the surface, which effectively alters their direction in the (ϑ, φ) -plane. As visible in Fig. 5b, c and discussed in the Supplementary Information, these distortions have a negligible effect in the present experiment, but do play a significant role for more complex surface charge distributions. For arbitrary charge distributions, however, the retrieval of the initial proton densities from the measured distribution is not straight forward.

To summarize, the physical meaning of the iterative optimization lies in its ability to retrieve the yield- and charge distribution on the surface of the nanoparticle from the measured momentum distribution, without a priori assumption about the process that generated them. We have added the following sentences to clarify this point.

The use of this iterative optimization procedure is not limited to the parameters of the present experiment. As shown in the SI, it provides a general framework for inverting the yield-to-momentum mapping and works for more complex cases, where the relation between the initial yield distribution and the final momentum distribution is much more intricate.

In addition, we have made a small change in Fig. 4, where the blue line now denotes a proton trajectory starting from the "north pole" of the particle, instead of the average proton trajectory. Comparing two trajectories with the same initial conditions seemed more consistent to us. We have adapted the caption accordingly:

The radial proton momentum (blue) is calculated for a proton released from the surface at the pole ($q=p/2$) of the particle.

References

- [1] M. Farnik, A. Pysanenko, K. Moriova, L. Ballauf, P. Scheier, J. Chalabala, and P. Slavicek, *Journal of Physical Chemistry A* 122, 8458 (2018).
- [2] S. Banerjee, J. Das, R. P. Alvarez, and S. Santra, *New J. Chem.* 34, 302 (2010).
- [3] A. Schweig and W. Thiel, *Mol. Phys.* 27, 265 (2006).
- [4] K. M. A. Refaey and W. A. Chupka, *J. Chem. Phys.* 48, 5205 (1968).
- [5] A. G. Sage, T. A. A. Oliver, R. N. Dixon, and M. N. R. Ashfold, *Mol. Phys.* 108, 945 (2010).
- [6] L. A. Curtiss, L. D. Kock, and J. A. Pople, *The Journal of Chemical Physics* 95, 4040 (1991).

Reviewers' comments:

Reviewer #1 (Remarks to the Author):

It has been revised extensively and now it is suitable for publication.

Reviewer #2 (Remarks to the Author):

I appreciate the authors' efforts to reply to my inquiries. However, their terminology, or usage of "near-field " still leads to considerable confusion, regarding my inquiry (1).

According to the conventional understanding for optical near fields, field enhancement near nanoparticle improves the event number, not energy supply for the event. That is why I asked whether they assume the nonlinear process, or so. I would ask the authors to clearly describe the mechanism to supply additional energy for dissociation.

Regarding question (2), I would like to comment that M^3C calculation only deals with the particle dynamics, not near field itself and the interaction with particles, which are critical for transient phenomena discussed by the authors.

Reviewer #1:

It has been revised extensively and now it is suitable for publication.

We would like to thank the reviewer for this evaluation and appreciate his support.

Reviewer #2 (Remarks to the Author):

I appreciate the authors' efforts to reply to my inquiries. However, their terminology, or usage of "near-field" still leads to considerable confusion, regarding my inquiry (1).

According to the conventional understanding for optical near fields, field enhancement near nanoparticle improves the event number, not energy supply for the event. That is why I asked whether they assume the nonlinear process, or so. I would ask the authors to clearly describe the mechanism to supply additional energy for dissociation.

The referee is right that in the regime of linear photoionization a larger local field strength only increases the number of ionization events compared to ionization in the pure laser field, but does not provide additional energy to each event (pure rate effect). This corresponds to the situation of Einstein's photo effect (single-photon process), where increasing the field strength only increases the number of emitted electrons but not their kinetic energy.

The situation is different in the nonlinear regime, where ionization is typically described in the multi-photon picture or in the quasistatic tunneling picture plus the subsequent (quantum) trajectory evolution. There, a larger field strength can lead to the deposition of more energy in the system (and in the emitted particle, say an electron), as the probability for the absorption of multiple photons increases nonlinearly and the ability to acquire drift momentum becomes dependent on the field strength. The latter aspect is the reason for the well-celebrated cut off scaling laws [1].

Moreover, if many electrons are liberated, the mutual interaction and collective response can also change the response qualitatively. Such behavior has also been found with the investigated nanospheres and is well-captured by our physical picture and modelling approach (see e.g. [4]).

The field induced ionization and dissociation processes considered in our study are all strongly nonlinear and require the absorption of multiple photons in the considered scenarios. This is why the near-field enhancement leads to both, a larger energy transfer and an event rate scaling that is nonlinearly in the field. To turn the argument around, the cross section for energy absorption changes, in contrast to the usual constant cross sections in the linear regime. The only thing that cannot be changed is the total energy content of the incident field.

Coming back to our scenario, we assume that the nonlinear formation process of H^+ is based on a multiphoton ionization of the adsorbed molecule followed by a dissociation process, which is triggered by rescattering electrons [2]. Neglecting local space charge fields from charge separation, the return energy of the rescattered electrons can reach 3.2 times the local ponderomotive energy, which scales linearly with the local intensity (see Figure below). This picture holds when the electron quiver amplitude is small compared to the $(1/e)$ -decay length of the optical field, i.e. when the field strength does not change significantly during the motion of the electron in the near-field. This is the case for the considered particle dimensions and wavelengths [3].

As an illustration of the spatial near-field distribution we added the Figure SI5 (see also answer (2)).

As a result, the field enhancement does lead to a higher impact energy of recollision electrons and can thus change the ionization and dissociation dynamics by more than a mere rate enhancement.

We hope that these arguments could resolve any possible remaining confusion about the role of enhancement.

Regarding question (2), I would like to comment that M³C calculation only deals with the particle dynamics, not near field itself and the interaction with particles, which are critical for transient phenomena discussed by the authors.

We have to disagree with the comment that the M³C simulations only deal with charged particle dynamics. In contrary, they also include the near-field around the nanoparticle using a two-level approach that accounts for dispersive electromagnetic field propagation in linear response (first layer) plus the self-consistent (!) mean-field generated by the free electrons and ions (second layer, quasistatic approximation) that result from (nonlinear) ionization in the near-field. The combination of both fields together with the collisional dynamics of the free electrons form the complete dynamics captured by the model. Hence, the near-field is the driver for the particle (electrons, ions) trajectories and the particle movement produces a feedback in the field. In the limit of relatively small ionization fractions the no-depletion approximation of the linear field contribution is well justified.

We are not completely sure what the referee meant by “particles” (electrons ions, or the nanoparticle itself) – we hope that our revised explanation clarifies all details. We therefore improved the Supporting Information in this regard. The added paragraph explains how the M³C calculations treat the near-field around the nanoparticle generated by the laser and how many-particle interactions are covered in the model.

THEORETICAL SIMULATIONS

A detailed description of the Mean-field Mie Monte-Carlo (M³C) calculations is given in Refs. [4,5] including the mathematical formulas describing the fields and the equations of motion. Below we summarize the main aspects of the M³C simulations:

(1) The linear-response contribution to the near-fields is obtained via a spectral decomposition of the incident laser pulses in combination with (frequency-dependent) spatial modes for the spherical geometry, that are calculated analytically utilizing the Mie solution of Maxwell's equations. The obtained spatially enhanced near-fields (see Fig. SI5) provide a proper description of the nanoparticle's electromagnetic response to the incident pulses for moderate intensities, at which the local polarization of bound electrons (dielectric medium) is mostly linear with respect to the applied electric fields. We therefore denoted this field as the (linear) near-field in the context of this work. We like to note that the Kerr-type non-linear bound state polarization is much smaller than the linear contribution in the considered intensity range.

(2) With increasing intensity, nonlinear effects start to become increasingly relevant due to stronger ionization of the target and the resulting generation of large numbers of free charges (residual ions at the nanoparticle surface and liberated electrons). The displacement of the free electrons from their host ions gives rise to an additional polarization that can exceed the bound state polarization and must be included (plasma response). We include both, the polarization of the sphere resulting from the free charges as well as the Coulomb interaction among the free charges, via a self-consistent mean field that we evaluate by solving Poisson's equation (using high order multipole expansion). This level, which is treated in quasistatic approximation, in principle supports extremely non-linear response behavior and can be seen as a full plasma model on top of the linear field description. As the liberated electrons are not removed from the model for the bound state polarization in point (1) above (this would break the linearity needed for the spectral decomposition), the relative ionization per atom should remain small. This is fulfilled in the scenarios treated in our study.

(3) At every time step, tunneling ionization is evaluated by Monte-Carlo methods and considering an atomic tunneling rate (ADK-rate) for the local intensity resulting from the combined linear near-field and the mean field. Upon successful ionization events electron trajectories are launched and propagated by integration of classical equations of motion in the combined near-fields.

The self-consistent treatment of the dynamics in the model stages (1)-(3) form the employed M³C description.

Figure SI5: Near-field distribution around the nanoparticle. The plot shows the field distribution around a 300 nm SiO₂ particle for a 720 nm wavelength laser field with an intensity of 4×10^{13} W/cm². The color denotes the field enhancement α relative to the incident field strength E_0 and the associated rescattering energy ($3.2 \times \alpha \times U_p$).

- [1] M. Busuladžić, A. Gazibegović-Busuladžić, and D. B. Milošević, *Laser Phys.* 16, 289 (2006).
- [2] P. B. Corkum, *Phys. Rev. Lett.* 71, 1994 (1993).
- [3] G. Herink, D. R. Solli, M. Gulde, and C. Ropers, *Nature* 483, 190 (2012).
- [4] F. Süßmann *et al.*, *Nat. Commun.* 6, 7944 (2015).
- [5] L. Seiffert, F. Süßmann, S. Zherebtsov, P. Rupp, C. Peltz, E. Rühl, M. F. Kling, and T. Fennel, *Appl. Phys. B* 122, 1 (2016).

Reviewers' comments:

Reviewer #2 (Remarks to the authors)

(1) This is one of what I have expected to have from the beginning, and conventional understanding of optical near-field phenomena. If the authors provide a typical number of field intensity enhancements, it would be helpful for the readers.

(2) On the basis of the above understanding, the phenomena claimed seem to be essentially nonlinear ones. Optical near fields are originated from the interactions between photons and electrons (holes) in the nanoparticles, then the framework of linear response theory that M3C is based on is not adequate to analyze the nonlinear ones.

Numerical explanation of the experimental results is qualitative, and thus the authors' emphasis on the completeness of the M3C theory is misleading. The reviewer recommends the authors to delete all such misleading terminology from the manuscript, and then the manuscript would be acceptable for the publication, assuming the experimental results have something new to give significant impact.

Reviewer #2:

(1) This is one of what I have expected to have from the beginning, and conventional understanding of optical near-field phenomena. If the authors provide a typical number of field intensity enhancements, it would be helpful for the readers.

We are pleased that the explanations in the Supporting Information could resolve possible misconceptions. To give a rough estimate we followed the reviewer's suggestion and added the following information:

The highest field enhancement due to Mie scattering is 1.8. The field is further enhanced by the self-consistent mean field (see Ref. [3]). The maximum of the total field enhancement arising from both the Mie field and the mean field amounts to 2.3 (relative to E_0).

(2a) On the basis of the above understanding, the phenomena claimed seem to be essentially nonlinear ones. Optical near fields are originated from the interactions between photons and electrons (holes) in the nanoparticles, then the framework of linear response theory that M3C is based on is not adequate to analyze the nonlinear ones.

We appreciate and fully agree with the statement of the reviewer regarding the nonlinear nature of the electron dynamics and we are happy that the discussion now makes this point clear. However, we sympathetically disagree with the claim that the framework of the utilized model is inadequate for describing this nonlinear electron dynamics. We like to explain this aspect in a bit more detail to clarify possible misunderstandings.

The actual local electric field (magnetic effects are negligible for the dynamics in our intensity regime) can, without loss of generality, be considered as a sum of a leading "linear" term and a correction that reflects the additional effect of nonlinear processes. This is a central paradigm of nonlinear optics. In our case the linear term is treated via the general solution for the electromagnetic field of a sphere – i.e. the Mie solution to Maxwell's equations. The additional term is more difficult to catch with a full electromagnetic treatment (e.g. as done with our MicPIC approach in Varin et al. Phys. Rev. Lett. 108:175007, 2012). A common approximation is the treatment of the nonlinear part in electrostatic approximation, i.e., via solving Poisson's equation for the charge density that results from the field driven motion of previously absent conduction electrons (free electrons and residual ions). This charge density is the one that deviates from the linear response polarization – this is what we do in our M3C model. We like to stress that not just the mere Coulomb interaction of the free charges is captured by our electrostatic solver, but also the associated induced polarization in the sphere. The employed approximation (electromagnetic linear field plus self-consistent electrostatic non-linear field from conduction electrons), which is described in great technical detail in the published dissertation of L. Seiffert (https://doi.org/10.18453/rosdok_id00002417), is based on two requirements:

(i) the nonlinear polarization of the bound electrons must be negligibly small

and

(ii) the field propagation effects resulting from the fields stemming from the nonlinear polarization must be unimportant

Both requirements are safely met in our case.

regarding (i):

As we have already tried to stress in our previous response, the nonlinear polarization of bound electrons is dominated by the Kerr term (Kerr constant for SiO₂ is roughly $n_2 = 3 \times 10^{-16} \text{ cm}^2/\text{W}$, see e.g. Boyd "Nonlinear Optics"), which modifies the local refractive index by less than a per cent for our field intensities. Second, the depletion of bound electrons via tunnel ionization also acts like a "non-linearity" – which, however, is also small in our case as the ionization degree is also far less than one per cent.

The dominating nonlinear polarization term results from the free motion of the liberated electrons – this is the leading term we describe and include self-consistently.

regarding (ii):

The plasma field stems from very localized charges (displacement of tunnel electrons from residual ions) - hence the propagation effect can be safely considered as a higher order correction that by no means will defeat or substantially modify the dynamics.

Hence, we cover the dominant factors (linear field plus nonlinear plasma field from tunnel electrons) in our effectively classical model. The depletion effect (we think this is what the reviewer refers to as holes) and the Kerr contributions are VERY small, such that we do not see why our approach should be inadequate.

We have added a link to the published dissertation as an in-depth technical reference that should remove all concerns regarding applicability of the method.

(2b) Numerical explanation of the experimental results is qualitative, and thus the authors' emphasis on the completeness of the M3C theory is misleading. The reviewer recommends the authors to delete all such misleading terminology from the manuscript, and then the manuscript would be acceptable for the publication, assuming the experimental results have something new to give significant impact.

We already emphasized the qualitative agreement – as suggested – in the following sections of the submitted manuscript:

- Line 152—155: The calculation results for 110 nm and 300 nm particles (Fig. 3b, d, and dashed lines in Fig. 3e-h) reproduce the characteristic trend of the experimental observations and in particular, the observed change in proton momentum distributions with nanoparticle size
- Line 157—158: Based on the good qualitative agreement, we use the simulations to disentangle the different effects leading to the observed momentum distributions.
- Line 252—254: Qualitative agreement with the experimental data is obtained from semi-classical simulations that incorporate the near-field, the rate of the dissociative ionization, and many-particle charge interactions.

In order to avoid any “misleading terminologies” we adapted the following sentences:

Line 243—244: The retrieved set of optimized parameters is in qualitative agreement with the charge distribution and dissociation rate calculated using full M3C-simulations

Line 212—214: In order to solve this inverse problem, we use our simulations to provide a more quantitative description of the mapping between the initial proton position and the final proton momentum.

Line 202—304: Without near-field enhancement, the maximal recollision energy is only about 9 eV and H⁺ formation is strongly suppressed, which is consistent with our data.

Reviewers' comments:

Reviewer #2 (Remarks to the Author):

The authors still seem to insist on the validity of M3C approach even in nonlinear optical near-field processes. They follow the conventional theoretical treatment based on the linear process's dominance, and thus the reviewer asks them to show which results come from the nonlinear process in simulation, comparing experimental data with the simulation results without nonlinear optical near field effects. The reviewer doubts if they can show explicit results revealing non-linear effects because they assume a linear response theory assuming the nonlinear effects are small at the beginning. They stress "without near-field enhancement, the maximal recollision energy is only about 9 eV and H⁺ formation is strongly suppressed," but no explanation that self-consistent M3C simulation incorporates the process obtaining the additional energy required. If my understanding is correct, optical near fields mainly originate from the interaction incident laser field and nanoparticles in this experimental situation. If the authors claim the validity of M3C simulation to interpret their experimental data, the reviewer asks them to show how energy gain processes are self-consistently included in M3C simulation and indicate which part of the momentum mapping(?) results mainly originate from nonlinear-dependent simulation.

Response to Reviewer #2:

The authors still seem to insist on the validity of M3C approach even in nonlinear optical near-field processes. They follow the conventional theoretical treatment based on the linear process's dominance, and thus the reviewer asks them to show which results come from the nonlinear process in simulation, comparing experimental data with the simulation results without nonlinear optical near field effects. The reviewer doubts if they can show explicit results revealing non-linear effects because they assume a linear response theory assuming the nonlinear effects are small at the beginning.

They stress "without near-field enhancement, the maximal recollision energy is only about 9 eV and H⁺ formation is strongly suppressed," but no explanation that self-consistent M3C simulation incorporates the process obtaining the additional energy required. If my understanding is correct, optical near fields mainly originate from the interaction incident laser field and nanoparticles in this experimental situation. If the authors claim the validity of M3C simulation to interpret their experimental data, the reviewer asks them to show how energy gain processes are self-consistently included in M3C simulation and indicate which part of the momentum mapping(?) results mainly originate from nonlinear-dependent simulation.

Let us clarify the relevant physics and the assumptions made in the model and their applicability at the experimental conditions. The physics governing the interaction of the incoming laser pulse with the nanoparticle can be separated into two distinct steps.

- (1) The (mostly linear) polarization of the nanoparticle generates optical near-fields.
- (2) The near-field itself, which is strongest at the surface, can drive local processes beyond linear polarization, such as tunneling or multiphoton ionization. Note that the resulting free electrons can be accelerated further by the local near field and may trigger secondary processes, as predicted by our simulations.

The fact that near-fields may be dominated by the linear response polarization does by no means exclude nonlinear processes in the field hot spots. We like to point out, that the rescattering of electrons from metallic nanotips was extremely successfully described in that way, i.e. with the nonlinear ionization and acceleration driven by linear near-fields. The near-field enhancement at the apex of these nanotips has enabled electron creation and acceleration at such low incident intensities, that the nonlinear emission could be driven with Ti:sapphire oscillators. This work (which is just one example of near-field driven strong-field processes) was published very prominently by e.g. the Hommelhoff and Ropers groups (M. Krüger, M. Schenk, and P. Hommelhoff, *Nature* 475, 78 (2011); G. Herink, D. R. Solli, M. Gulde, and C. Ropers, *Nature* 483, 190 (2012)).

Our M³C treatment even goes beyond as it incorporates the linear polarization (as in Hommelhoff's and Roper's work) and the self-consistent classical plasma dynamics that builds up if charges are set free, e.g. by tunneling in a field hot spot.

The concern of the referee can thus only stem from the neglect of the nonlinear bound state polarization. In our previous response, we have already put forward arguments, why this contribution is small/negligible in our scenario. However, we have not yet supported this

conclusion by a quantitative number or ratio of the effect on the near field. We thus incorporated such last missing piece of evidence in the revised supplement.

The maximum intensity on the surface is about $2.6 \times 10^{14} \text{W/cm}^2$, while inside the particle the field is screened and reaches up to $1.25 \times 10^{14} \text{W/cm}^2$. The Kerr effect at this intensity results in a change of refractive index by $\Delta n = n_2^* I = 0.037$. When taking this into account in the computation of the near-field, the resulting change of maximum field enhancement is below 5%. We like to point out here that such a small change results in practically the same maximum energies reachable for near-field driven rescattering electrons, see figure R1 below.

Figure R1: Near-field distribution around the nanoparticle. The plot shows the field distribution around a 300 nm SiO₂ particle for a 720 nm laser field. The left and right plots are calculated for a refractive index of 1.456 and 1.494, respectively, corresponding to the cases of near-fields without and with Kerr nonlinearity taken into account.

We agree with the referee that this is worth mentioning in the manuscript and we have included the following sentence in the Supporting Information to clarify the impact of taking into account the Kerr nonlinearity in the near-field description:

“Possible deviations in the near-field enhancement due to the Kerr-effect are estimated to be well below 5% for the considered experimental conditions.”

Regarding the reviewer’s concern about the recollision energy, classically the maximum value for ponderomotive acceleration can be calculated as: $E_{kin} = 3.2 U_p$ [1]. In enhanced near-fields the maximum ponderomotive energy can be obtained from the field enhancement a simply as: $E_{kin} = a^2 3.2 U_p$. The resulting numbers using these formulas derived from the near-fields are displayed in figure R1 (see right labels on the color scale), with and without taking the non-linearity of the near-fields in terms of the Kerr nonlinearity into account. The main differences in our M³C simulations (cf. Fig. R2) with respect to these simple estimates are stemming from:

- i) Taking into account the actual local time-dependent Mie-field for the electron acceleration in a self-consistent manner, which for the particles studied in our work yields essentially identical results to the simple formula above.

- ii) Taking into account the charge interaction, which in fact only to a minor extent affects the recollision energy as already investigated in detail in Ref. [2], and as can be seen in Fig. R2 blue line vs. red line.

We can thus conclude that the recollision spectra (cf. Fig. R2) from our M³C simulations are consistent with expectations based on simple classical arguments.

Figure R2: Electron recollision energies for 300nm silica particles and an intensity of $3 \times 10^{13} \text{W/cm}^2$. The full M³C simulation includes the linear near-fields and the mean-field, while the “no mean-field” simulation neglects the mean-field contribution. The dashed lines indicate the classical recollision energies. The variable α is the near-field enhancement factor, which amounts to 1.8 in this case.

We finally like to point out that the momentum mapping is a result of the created charges, repelling the created molecular fragment from the nanoparticle surface. The charges are a result of a nonlinear interaction with the fields and are fully taken into account in the M³C model. The accelerating fields at each point in space and time for emitted ions result from a combination of the Mie field and charged particle mean-field. Without this charge interaction, H⁺ ions on the surface of the nanoparticles would gain less than 0.5 a.u. in momentum in the laser field, while this is around 70 a.u. with the charge interaction taken into account (and in good agreement with the experimental values), see Fig. R3. We can thus conclude that nonlinear processes are at the heart of the momentum mapping, and this is reproduced even quantitatively (as evident from similar maxima momenta of the ions) in the M³C simulations.

Figure R3: Left: Typical ion trajectories for 300nm silica particles in 3×10^{13} W/cm² laser fields. The results from the full simulations are described in detail in Fig. 4 in the manuscript. The “no interaction” case neglects the charge interaction and only considers the laser fields including the linear near-field. Right: Enlarged view of the average ion trajectory (thick solid line) and variation in individual trajectories (shaded region).

- [1] P. B. Corkum, Phys. Rev. Lett. 71, 1994 (1993).
- [2] L. Seiffert, F. Süßmann, S. Zherebtsov, P. Rupp, C. Peltz, E. Rühl, M. F. Kling, and T. Fennel, Appl. Phys. B 122, 1 (2016).

REVIEWERS' COMMENTS:

Reviewer #2 (Remarks to the Author):

I still feel there is a gap between us. The point of the argument is as follows:

Whether are the nonlinear near-field effects self-consistently included in M3C simulation, from the theoretical viewpoint?

According to the reference (2) the authors cited in the previous reply report, classical Maxwell em. fields are described in terms of Mie scattering that is believed not valid for small particles with a size less than $\lambda/2$, where dynamical field fluctuation is critical and ϵ_{mie} varies in space and time. Thus it is not constant nor Mie fields and mean fields decoupled, as assumed in the reference.

On the other hand, macroscopic Maxwell em. fields are averaged in space and time to suitably describe far-field phenomena in the region larger than λ . That is why the reference (2) assumed the separation of dominant free carrier's polarization from the near-field one.

Actually, they have treated free carriers dynamics in a Coulomb potential self-consistently, in a sense of Hartree-Fock approximation, not including near fields around the small particles induced by the incident laser field that should influence the tunneling of electrons, just assumed without theoretical derivation of the relation between the induced near fields and electron tunneling.

In my opinion, M3C theory just provides a nonlinear description based on a H-F like Coulomb potential for free carriers generated, without any self-consistent theoretical description of near fields induced by the incident laser. Therefore it seems that the current authors greatly exaggerate the self-consistent near-field effect of M3C in the analysis of experimental results.

For example, they claim good agreements between (a) and (b), (c) and (d) in Fig.3, but polarization behavior in the regions ($\theta \neq 0$, $\varphi \neq 0$) seems different completely.

Reviewer #2 (Remarks to the Author):

I still feel there is a gap between us. The point of the argument is as follows:

Our response:

We share the desire of the reviewer to employ the most reasonable description of the physics. However, before discussing specific points, the reviewer will also agree that compromises are needed. A fully quantum mechanical description including all electrons and nuclei is impossible for the problem of interest. Moreover, if also the field would be treated quantum mechanically, the treatment will get even more pathological from the practical side and, for complex scenarios as ours, most likely not very insightful. Hence, strong approximations are inevitable to explore and understand the physics. The experimental evidence that a size dependent deformation of the angular fragment distribution takes place clearly calls for a modelling that includes field propagation. It should be noted, that field propagation is typically not included in most first principle simulations a la TDDFT, which may include electronic dynamics at much higher level of sophistication but will miss the propagation effect completely. The methodology employed in the M3C model is driven by the quest to understand and include the dominant processes in a reasonable and tractable form – not more but also not less. With this in mind, we like to address the individual concerns mentioned by the reviewer in more detail below.

Whether are the nonlinear near-field effects self-consistently included in M3C simulation, from the theoretical viewpoint?

According to the reference (2) the authors cited in the previous reply report, classical Maxwell em. fields are described in terms of Mie scattering that is believed not valid for small particles with a size less than $\lambda/2$, where dynamical field fluctuation is critical and ϵ_{mie} varies in space and time. Thus it is not constant nor Mie fields and mean fields decoupled, as assumed in the reference.

We sympathetically disagree with the implicit statement that a description of optical phenomena like near-fields driven dynamics and scattering based on classical continuum electromagnetic theory (i.e. Maxwell's equations, like Mie solution for spheres) is per se not valid at scales below $\lambda/2$. The continuum description blurs out the local field effects on the scale of interatomic distances, but certainly not misses the physics at the scale of the wavelength. If this would be true, the treatment of the transmission and reflection in e.g. films much thinner than the wavelength by continuum Maxwell equations should be wrong as well. Likewise, all published work on near-field induced electron dynamics should be wrong too. We have serious problems to follow the argument of inapplicability of classical Maxwell theory (Mie solution for our geometry).

Regarding fluctuations, we like to argue that we consider a high intensity scenario for short time scales, where quantum field fluctuations, spontaneous emission etc. can be safely neglected.

Next, if classical continuum em. theory is employed and averaging over fine details at interatomic scales is agreed on, there is infinite freedom to split the currents and polarization in different terms. An often most convenient choice is the splitting into the bound and free carrier motion – which puts no restrictions on the physics yet. Further, it is important to stress that we never claimed that the resulting differential equations for bound and free carrier motion decouple. In fact, laser-induced Mie contributions to the fields drive the free carriers. The field from the free carrier charge density in turn creates additional polarization of bound electrons, providing a feedback that, at the end of the days, would be analogous to a modified permittivity. In our model we treat the central part of the bound state polarization (the part driven by the laser) with full retardation but in linear approximation. As explained in the text, the polarization response that follows the local field non-

linearly (lead by Kerr term) and the depletion term (due to ionization) are neglected, which is justified as they can be estimated to be small. The couplings between free-carrier-induced polarization and the free carriers as well as the free carrier-free carrier interaction are included on the mean field level.

In recognition of the implications of the averaging over interatomic scales and the neglect of bound state non-linearity, we are still confident that there is no fundamental conceptual flaw in the dynamical model. It is understood that approximations need to be stated – but this is done in an honest and clear way. We are fully with the reviewer that it would be interesting to study the impact of sub-interatomic scale effects and higher order susceptibility contributions that are missed by our approximations, but these effects will certainly not erase or qualitatively change the angular directionality features in our parameter range.

On the other hand, macroscopic Maxwell em. fields are averaged in space and time to suitably describe far-field phenomena in the region larger than λ . That is why the reference (2) assumed the separation of dominant free carrier's polarization from the near-field one.

In our model we consider a time-resolved propagation of fields and carriers – the only averaging performed in addition to the continuum picture is the mean field averaging over trajectories of free carriers, equivalent to the removal of electron-electron collisions (dynamical correlations). We do not perform a temporal averaging in the propagation. Most importantly, as argued in detail above, the fields are sampled and physically meaningful (in the framework of the employed approximation) on scales much smaller than the wavelength.

Actually, they have treated free carriers dynamics in a Coulomb potential self-consistently, in a sense of Hartree-Fock approximation, not including near fields around the small particles induced by the incident laser field that should influence the tunneling of electrons, just assumed without theoretical derivation of the relation between the induced near fields and electron tunneling.

We like to point out that we use the total local field as predicted by solving Maxwell's equations (with the mentioned approximations), including (i) the laser, (ii) the polarization field induced by the laser itself, (iii) the free carrier field, and (iv) the polarization field induced by the free-carriers including the resulting dynamical feedback (this requires the self-consistent solution). We then use this local field to drive carriers and the ionization process, the latter approximated by a simple but widely used ionization model. The underlying logic is that the ionization rate just requires knowledge of the local field, irrespective of the particular nature of the individual contributions (i-iv). Hence, we already do what the reviewer is suggesting, i.e.: we include the near fields around the small particles induced by the incident laser field plus the free-carrier induced modifications in the description of the tunnelling of electrons.

In my opinion, M3C theory just provides a nonlinear description based on a H-F like Coulomb potential for free carriers generated, without any self-consistent theoretical description of near fields induced by the incident laser. Therefore it seems that the current authors greatly exaggerate the self-consistent near-field effect of M3C in the analysis of experimental results.

Because of the neglect of Kerr terms in our model, the response remains linear if there is no ionization. As soon as ionization starts, the feedback from the free carrier dynamics will lead to a near-field that deviates from the linear Mie result because of the additional free carrier field itself and because of the additional polarization induced by the free carrier field. By construction of the model, we split the total field into the linear part and a correction to it. It is this correction, which is treated in quasistatic mean field approximation, that accommodates the self-consistent linear and non-linear feedback also for the laser-induced near-field (this is what the reviewer has questioned).

The conceptual splitting into a linear and a correction term by no means corresponds to a neglect of a laser-induced non-linear response resulting from the free carriers. If both the linear and correction terms would be treated with retardation, the result would be identical to a full solution without splitting. In our treatment, the correction (including the non-linear feedback) is instantaneous and quasistatic – but it is for sure still self-consistent. As we clearly state the quasistatic approximation in the correction term (mean field term) in the manuscript, we cannot see how our description, claims, or the actual treatment “greatly exaggerate” the near field effect. The splitting into a simple linear and a complicated non-linear term where the latter is treated using stronger approximations has proven to be very successful and is very common in non-linear optics.

If readers want to convince themselves of the approximations, logic, and steps required to arrive at the employed splitting, the Supplement included a reference to the full derivation of the M3C core model (the PhD-thesis of L. Seiffert, published online). We like to note that this thesis was suggested for distinction by the renowned theorist in attosecond theory – Misha Yu Ivanov. We are confident that our approach is conceptually sound in the framework of the explicitly spelled-out approximations and does provide what we are after – a tractable model that is employed to explore the main physics. The fact that the above mentioned main trends are captured supports that the model includes the main physics.

For example, they claim good agreements between (a) and (b), (c) and (d) in Fig.3, but polarization behavior in the regions ($\theta \neq 0$, $\phi \neq 0$) seems different completely.

The main feature of the data displayed in Fig. 3 is the angular emission with hot spots that are at the poles for small spheres and are tilted in the propagation direction for larger spheres. These signatures are clearly captured by the simulation. Let us look again at the statement we made in the text:

“The calculation results for 110 nm and 300 nm particles (Fig. 3b, d, and dashed lines in Fig. 3e-h) reproduce the characteristic trend of the experimental observations and in particular, the observed change in proton momentum distributions with nanoparticle size.”

We claimed the reproduction of the main trends, including a shift of the distribution, which still appears justified, honest, and correct to us. We updated the corresponding text to express now even more clearly which central feature we have in mind when discussing the figure. In order to avoid any possible misunderstanding, we rephrased the sentence in the following way:

“The calculation results for 110 nm and 300 nm particles (Fig. 3b, d, and dashed lines in Fig. 3e-h) reproduce the characteristic trend of the experimental observations, i.e. the presence of pronounced directional emission hot spots and their movement in propagation direction towards the back side of the particle with increasing diameter. This trend is expressed most clearly in the peak shift of the projected proton momentum distributions.”

To address the reviewer’s concerns to a satisfactory extent we applied the following changes:

Main text

Line 30: added “qualitatively”

Line 146: changed to “explore”

Line 149—153: changed to “In our model, electrons are liberated via tunnel ionization, under the action of the total electric field consisting of the laser field and the induced field of the sphere resulting from bound and free charges. The field is described using a self-consistent two-level scheme, where the linear contribution is treated via the Mie solution and the correction that includes all non-linear contributions is described in quasistatic mean field approximation.”

The captions of Figs. 3 and 5 now contain more information about the definition of the spherical coordinates θ and φ .

Supplementary Information

We added the definition for the spherical coordinates in the Supplementary Note 1

We rephrased section (2) of the Supplementary Notes 3.